# The Fibro-Inflammatory Response in the Glaucomatous Optic Nerve Head

**DOI:** 10.3390/ijms241713240

**Published:** 2023-08-26

**Authors:** Emma K. Geiduschek, Colleen M. McDowell

**Affiliations:** Department of Ophthalmology and Visual Sciences, University of Wisconsin-Madison, Madison, WI 53705, USA

**Keywords:** glaucoma, TLR4, fibrosis, immune response, optic nerve head, TGFβ2

## Abstract

Glaucoma is a progressive disease and the leading cause of irreversible blindness. The limited therapeutics available are only able to manage the common risk factor of glaucoma, elevated intraocular pressure (IOP), indicating a great need for understanding the cellular mechanisms behind optic nerve head (ONH) damage during disease progression. Here we review the known inflammatory and fibrotic changes occurring in the ONH. In addition, we describe a novel mechanism of toll-like receptor 4 (TLR4) and transforming growth factor beta-2 (TGFβ2) signaling crosstalk in the cells of the ONH that contribute to glaucomatous damage. Understanding molecular signaling within and between the cells of the ONH can help identify new drug targets and therapeutics.

## 1. Introduction

Glaucoma is currently a leading cause of irreversible blindness, estimated to effect 75 million individuals worldwide, which figure is proposed to increase to over 100 million by the year 2040 [1,2]. The glaucomas are a heterogenous group of optic neuropathies characterized by the loss of retinal ganglion cells (RGCs) and subsequent optic nerve head damage (ONH) and changes, resulting in a progressive loss of vision in distinctive and well-studied patterns [3,4,5]. In this review, we are focusing on the most common form of glaucoma, primary open-angle glaucoma (POAG). Risk factors include age, race, and sex, though much of the attention is focused on the role of intraocular pressure (IOP) and IOP management due to the high correlation between increased IOP and decreasing vision scores [6,7]. Elevated IOP has been implicated as the most prominent risk factor for the development and progression of glaucoma, and it has been shown by multiple groups across different populations that lowering IOP through medication or surgery can delay glaucoma progression [7,8,9]. IOP homeostasis is disrupted when the production of aqueous humor (AH) is not balanced by the rate of AH drainage through the outflow pathways in the iridocorneal angle of the eye. Most of the AH outflow is through the trabecular meshwork (TM) and Schlemm’s canal, where extracellular matrix (ECM) proteins form a fluid-flow pathway for the AH [10]. In glaucoma, increased resistance through the outflow pathways in the TM, particularly in the juxtacanalicular connective tissue (JCT) region and at the inner wall of Schlemm’s canal, results in increases in IOP [11]. Unfortunately, even with well-managed IOP through pharmaceuticals or surgery, many patients still exhibit progressive vision loss [12]. Exploring the molecular and cellular mechanisms behind glaucoma progression at the ONH will help address the crucial need for more effective treatments.

Glaucoma is defined by the loss of retinal ganglion cells (RGCs), the thinning of the retinal nerve fiber layer, and the cupping and remodeling of the ONH resulting in a gradual loss of vision [13,14]. Multiple insults, including chronic mechanical stress due to high IOP, hypoxia microenvironments, and loss of neurotrophic factors and nutrient transport, all contribute to the loss of the RGCs [15,16]. Damage to the exiting RGC axons is most vulnerable at the ONH due to the 90° turn the axons make exiting the ONH, which occurs at the layer of the lamina cribrosa (LC), a mesh-like connective tissue structure of pores through which the RGC axons travel to the eye [13,17]. The LC region of the ONH acts as both a physical support to these exiting RGC axons, as well as the scaffolding for support cells to deliver nutrients and survey the microenvironment for potential cites of damage. The biomechanics of the LC region are highly implicated in glaucoma pathology, including balancing forces between IOP and intracranial pressure, the stiffness and elasticity of the LC region, and tissue-specific biomechanical responses [18]. These biomechanical changes directly interfere with the interaction between the LC scaffolding and ECM proteins, the RGC axons, and the supporting cells in the ONH.

There are three major supporting cell types in the LC region: Iba1 positive microglial cells, glial fibrillary acidic protein (GFAP) positive ONH astrocytes, and α-smooth muscle actin (αSMA) positive LC cells. Microglia, the resident macrophages of the immune system, are regularly spaced along the walls of the blood vessels, within the glial columns, and in the LC region in the ONH to optimize their ability to survey their microenvironment [19,20,21]. The astrocytes are located both along the longitudinal LC beams and in a transverse orientation across multiple beams [22]. Finally, LC cells are located within the LC beams, within or between the cribriform plates [22]. All three of these cell types have been implicated in glaucomatous pathophysiology as described below. 

In homeostatic environments, microglia efficiently clear dead cells and cellular debris [23]. This “resting” state is a highly active process of constantly surveying the microenvironment [20]. Upon activation during disease states or with damage to the CNS, microglia undergo morphological changes from ramified surveyors to an ameboid shape [24,25,26], and rapidly respond and migrate to the site of damage within minutes [21,27]. This migration is ATP-dependent, where ATP activates the P2Y12 receptor on the microglia [28]. Microgliosis is known to be associated with many CNS diseases including Parkinson’s disease, Alzheimer’s, and glaucoma [20]. Increased numbers of activated microglia are seen in the human ONH in glaucoma compared to age-matched controls [19,29]. In mouse models of glaucoma, there is an increased number of microglia and increased activation of microglia prior to RGC death and axonal damage [30,31,32]. In addition, the severity of early microglial activation correlates with the severity of RGC and ON axon pathology [32]. Genes that are expressed in activated microglia are significantly increased in mouse and rat models of glaucoma, such as: major histocompatibility complex–II (MHC-II), highly involved in the adaptive immune response; complement 1 complex components, involved in the innate immune response; P2Y12, the receptor responsible for initiating microglial migration to sites of damage; and TLR4, also involved in the innate immune response [33,34,35]. Minocycline, an inhibitor of microglial activation, has been previously used to explore the role of microglia in glaucoma progression. Minocycline treatment has been shown to enhance the survival of RGCs and rescue RGC nutrient transport [36,37,38]. Driving the immune response in human and mouse models of glaucoma, microglia sit as a key mediator for the progressive pathophysiology in the disease. 

Astrocytes are the most common glial cell in the mammalian ONH [22], providing cellular support by facilitating nutrient transport and distribution throughout the ONH as well as secreting ECM proteins to provide physical structure support [22,39]. Human ONH astrocytes have region-dependent molecular heterogeneity [40]. There are three subtypes of astrocytes in the ONH: Type 1A, type 1B, and type 2. Type 1A are in the unmyelinated LC and prelaminar regions of the ONH and express GFAP, but are negative for neural cell adhesion molecule (NCAM). Type 1B are also in the LC and prelaminar regions, but express both GFAP and NCAM. Type 2 astrocytes are in the myelinated post-laminar region of the ONH [41]. In early glaucoma, ONH astrocytes are hypothesized to be protective. It is known that astrocytes can redistribute nutrients from healthy to stressed microenvironments after IOP increases, and the knock-out of astrocyte reactivity genes early in a disease model results in more RGC death [39,42,43]. In later glaucoma disease states, astrocytes transition to a neurotoxic phenotype, a process called astrogliosis. This reactive phenotype transition is hypothesized to be initiated by activated microglia and is microglia dependent, where mice without functioning microglia do not exhibit a reactive astrocyte phenotype after insult [44,45]. Reactive astrocytes in the LC region take on an activated physical phenotype, showing rounded bodies with a loss of cell processes [44,45,46]. Activated astrocytes in the glaucomatous ONH secrete higher levels of ECM proteins prominent in the LC region [47], and interfere with the exiting RGC anterograde transport of nutrients [39]. As regards glaucoma progression, increased IOP is known to result in a loss of nutrient transport and astrogliosis, implicating astrocytes as a major contributor to RGC axon damage and eventual progressive vision loss.

The main function of the LC cells is to secret ECM proteins such as collagens, elastin, and fibronectin, to maintain the structural laminar beams that physically support the exiting RGC axons [48]. LC cells are highly responsive to chemical and mechanical stimuli, altering their gene expression levels when exposed to transforming growth factors (TGFs), known to be involved in ocular wound healing and glaucoma pathophysiology, or under mechanical strain [49,50,51,52,53]. Previous studies have shown that LC cells collected from patients with POAG have upregulated ECM protein expression [54], and undergo fibrosis and mechanical failure compared to age-matched controls [55]. Glaucoma-like stimuli (TGFβ exposure, mechanical stress, hypoxia) have all generated increases in ECM proteins in LC cells [50,51,56,57], and increased immunostaining for enzymes controlling the breakdown of collagen and fibronectin that have been shown in the LC region of the ONH [29,49,58]. These data implicate a critical role for LC cells in glaucoma by contributing to the increased fibrosis and remodeling of the ONH.

All three supporting cell types in the ONH have been implicated in perpetuating glaucomatous damage (Table 1). However, the cell–cell signaling between these cell types and the RGC axons is not fully understood.

## 2. Fibrosis in the Glaucomatous ONH

The drastic changes to the ECM and increased fibrosis in the glaucomatous ONH and LC region have been extensively reviewed [14,49,70,71]. Fibrosis is defined as the excessive production and accumulation of ECM proteins, inducing structural and functional abnormalities in the affected tissue [72]. Elevated IOP causes significant strain and stress on the ONH region, resulting in posterior migration of the LC and eventual cupping of the ONH [22]. This mechanical strain induces increased fibrosis, specifically the deposition and dysregulation of ECM proteins elastin, tenascin, collagens I, IV, V, XI, proteoglycan, and fibronectin [22,54,55,56,73,74,75,76]. ECM deposition and dysregulation induces a plethora of physical changes, including elastosis, increased fibrosis, the thickening of the connective tissue around the ON fibers impairing nutrient transport, and disorganization of the regular collagen structure [22,55,73]. This ECM remodeling adversely affects the capacity of the LC to support the exiting RGC axons, predisposing them to the axonal compression and disruption of nutrient transport [39]. While it is established that elevated IOP leads to stress and strain on the ONH, the pathogenic molecular mechanisms responsible for the structural changes are not well understood.

One predominant hypothesis indicates TGFβ2-induced ECM synthesis as a major player in instigating and exacerbating the increased ECM buildup and dysregulation in glaucoma [64]. The molecular signaling pathway of TGFβ2-induced ECM synthesis has been well studied. TGFβ2 binding induces a heterotetrameric complex between two type I receptors and two type II receptors to initiate the canonical Smad signaling pathway [77]. The Smad signaling pathway results in the phosphorylation of Smad2/3, which colocalizes with Co-Smad4 in the nucleus of both human ONH astrocytes and LC cells [65,78]. The downregulation of Smad signaling is induced by the increased expression of SMAD7, which recruits Smad6 to inhibit the phosphorylation of SMAD2/3 by directly interacting with the TGFβ receptors intracellularly [79,80,81]. The Smad7 inhibition of TGFβ2-signaling is amplified by BMP and activin membrane-bound inhibitor (BAMBI). BAMBI cooperates with inhibitory Smad7 to prevent Smad3 phosphorylation [82]. BAMBI also acts as a pseudoreceptor by replacing one of the TGFβ-receptors in the receptor complex, preventing the phosphorylation of Smad2/3 [82]. This highly regulated TGFβ2 signaling pathway has been well studied in LC cells and tissues as well as in the ONH astrocytes.

TGFβ2 is the predominant isoform in the eye, and is found in in the trabecular meshwork (TM), aqueous humor, vitreous humor, neural retina, retinal pigment epithelium, and the ONH [64]. In the TM, TM cells are known to secret TGFβ2, and express TGFβ receptors and significantly increase ECM production in the presence of TGFβ2, indicating that the predominant outflow pathway for AH is under the influence of TGFβ signaling [83]. In the ONH, TGFβ2 levels are minor to non-existent in the healthy ONH [62], but have been shown to be increased 70–100-fold in the glaucomatous ONH compared to healthy age-matched controls, with staining primarily occurring in astrocytes (Table 1) [62]. The in vitro TGFβ2 treatment of primary human ONH astrocytes and LC cells induces the increased ECM deposition of elastin, collagen-IV, and fibronectin (FN) via canonical Smad signaling (Table 1) [56,62,63,64,65,84]. TGFβ2 is also present in activated microglia in the retinal nerve fiber layer, prelaminar, LC, and post-laminar regions in human glaucomatous ONHs [29]. Taken together, TGFβ2 signaling has the ability to interact with and influence both the front and back of the eye in glaucoma disease progression.

## 3. Inflammation in Glaucoma

Notably, a physiological level of inflammation is beneficial and necessary to fight infection, maintain tissue homeostasis, and recruit immune cells to clear sites of tissue damage [85]. However, when tissue is exposed to severe or prolonged levels of stress, inflammation plays a neurotoxic and deleterious role. It has been postulated that a prolonged exposure to mechanical stress and strain from elevated IOP, the subsequent loss of nutrients, and resulting hypoxic microenvironments, can transition the innate immune system in the ONH from protective to neurotoxic [86].

The role of innate immune activation and induced pathophysiology during glaucoma disease progression has been extensively reviewed [16,85,87], with ONH astrocytes [88], microglia [61,89], and LC cells implicated in initiating and responding to increased immune activation [66]. The ONH is an immune-privileged tissue, thus the defense systems consist of glial cells and the complement system [85]. These glial cells are the microglia and astrocytes, both found to be profoundly responsive to stress in the glaucomatous ONH. In a resting state, microglia survey their microenvironment and release neurotrophic factors to maintain RGC health [90]. Increased microglial activation is associated with human glaucoma [19,29], as well as in animal models of glaucoma [89,91]. As discussed earlier, early microglial activation predicates RGC damage and correlates with RGC degeneration severity [32,89], and induces a neurotoxic phenotype in astrocytes [44], which subsequently secrete pro-inflammatory and pro-fibrotic signals. These signals activate the innate immune signaling pathway by binding to toll-like receptors (TLRs). The constant glial-inflammatory response has recently been recognized as a crucial mechanism of the gradual neurodegeneration of the exiting RGCs [85,92]. Here, we will be focusing on the role of toll-like receptor 4 (TLR4) in the glaucomatous ONH.

The TLR family consists of 10 members (TLR1-10) in humans and 12 members (TLR1-9, TLR11-13) in mice [93]. The role of TLRs in glaucoma, age-related macular degeneration, and other retinal diseases has been extensively reviewed [93,94,95]. In this review we will be focusing on the role of TLR4 signaling in the glaucomatous ONH. TLR4 was first identified as the receptor for lipopolysaccharide, which is found on almost all Gram-negative bacteria and acts as an innate immunity signal [96,97]. Recent evidence has implicated TLR4 signaling in augmenting fibrosis and the production and regulation of ECM proteins in fibrotic diseases [31,83,98,99,100,101]. Importantly, *Tlr4* gene polymorphisms are associated with primary open angle glaucoma in multiple patient populations [102,103,104], and TLR4 pathway-related genes are differentially expressed in the retina and ONH of glaucomatous patients versus healthy patients [34,105]. TLR4 activation through the Myd88 signaling pathway increases the production of nuclear factor κ B (NFκB), which translocates to the nucleus to act as a transcription factor, initiating the production of pro-inflammatory signals as well as pro-fibrotic signals. Some of these inflammatory and fibrotic products can act as endogenous ligands for TLR4, known as damage-associated molecular patterns (DAMPs). TLR4 can then be again activated by these endogenous DAMPs, creating a positive feedback loop, leading to a progressive fibroinflammatory response [106].

DAMPs serve as key signals for tissue injury or damage. DAMPs are generated in situ in response to injury, oxidative stress, cell damage, or ECM remodeling [107]. Heat shock protein 60 was the first discovered endogenous DAMP, shown to induce cytokine synthesis through TLR4 activation [108]. Since then, dozens of endogenous DAMPs have been discovered, including different peptides, fatty acids, proteoglycans, and nucleic acids. The role of DAMPs and their involvement in immune system activation has been extensively reviewed by Piccinini and Midwood [107]; here, I will be discussing DAMPs and their involvement in primary open angle glaucomatous pathology. Interestingly, TLR4 can be activated by endogenous DAMPs that are known ECM molecules, including biglycan, tenascin-C, and the fibronectin EDA isoform (FN+EDA) [107]. Along with TLR4 expression differentiation [34,105], DAMPs such as tenascin-C and FN+EDA have also been identified as differentially expressed in the ONH and retina in glaucoma [34,56,75].

Biglycan is a proteoglycan that primarily supports tissue when exposed to compressional forces [109], such as the force on the ONH. Biglycan, like fibronectin, can be upregulated by TGFβ stimulation in renal cell cultures and is involved in the pathophysiology of several renal fibrosis disorders [110]. Similarly, biglycan has also been shown to be upregulated in cultured LC cells after mechanical stress [51]. Biglycan is released from the ECM in stressed tissues, interacting with ECM proteins COLI, II, III, IV, and elastin [111]. Crucially, biglycan is a potent pro-inflammatory signal that is known to activate TLR4 [112]. Biglycan knock-out mice live longer after LPS-induced sepsis than wild-type controls, and produce significantly lower levels of pro-inflammatory cytokines [112]. Although biglycan is known to be expressed in the LC region of the ONH, and altered by mechanical strain in LC cell cultures, it remains to be determined if expression levels differ between healthy and glaucomatous ONH in patients [113]. Additional studies are needed to fully understand the role of this important DAMP in TLR4-activation in the ONH.

Tenascin-C is a large glycoprotein expressed in neural and non-neural tissues [113], and is known to be expressed in the LC region of the ONH [113]. Importantly, tenascin-C levels are prominent in the LC region of the ONH in elderly donor eyes, implicating increased levels with age, a potent risk factor for glaucoma development [113]. In addition, tenascin-C has been shown to be a TLR4 activator, and maintains pro-inflammatory signaling in other immune-dependent diseases [69]. Tenascin-C protein expression is significantly increased in human and porcine TM cells exposed to high IOP using perfusion organ cultures [114]. In autoimmune glaucoma mouse models, increased levels of tenascin-C are found in both the retina and ONH [115]. In a rat model of glaucoma, tenascin-C mRNA was significantly increased in the ONH with early ON damage, and remained significantly elevated throughout the glaucoma progression compared to controls [35]. Similarly, the knock-out of tenascin-C lowered levels of reactive astrocytes in the ONH and reactive microglia in the mouse retina [116], highlighting the importance of this DAMP in disease progression. Functionally, tenascin-C is known to regulate TGFβ signaling during wound healing [117], an important pathway involved in the changes to the ONH and LC in glaucoma, as we discussed above. Importantly, tenascin-C protein expression is increased in the glaucomatous LC region of the ONH compared to healthy, age-matched controls [75], suggesting this DAMP may be intimately involved in the development of glaucomatous damage.

Finally, FN is a component of the ECM in the ONH, helping to form the intricate mesh-like layer of the LC region along with other ECM proteins [56]. FN is composed of either cellular FN (cFN) or plasma FN (pFN) [118]. cFN, found in the pericellular matrix, can contain various splice variant combinations of the extra domain—A (EDA), extra domain—B (EDB), or Type III homologies [118]. Conversely, pFN, secreted by hepatocytes directly into blood circulation, does not contain the EDA or EDB domains [118]. The FN+EDA isoform is a known DAMP that binds and activates TLR4 [56,83,119]. During embryonic development, the FN+EDA isoform is abundant, lowering to minimal levels in adult tissues except during tissue injury, repair, or disease states, where expression is again upregulated [120,121,122,123,124]. FN+EDA is increased in other fibrotic and immune diseases such as atherosclerosis, psoriasis, scleroderma, and rheumatoid arthritis, and in human glaucomatous TM tissue compared to healthy controls [83,125,126,127,128]. FN+EDA amplifies the TGFβ2-dependent ECM response in primary TM cells and can induce ocular hypertension in mouse models [83,101,129]. Importantly, we recently reported that FN+EDA is elevated in the LC region of the human glaucomatous ONH compared to healthy controls and amplifies the TGFβ2-dependent response in primary human LC cell cultures [56]. These data implicate FN+EDA as an important DAMP involved in modulating the glaucomatous ONH.

Here, we have described the role of ECM DAMPs in both immune system activation and glaucoma pathophysiology. Increased levels of DAMPs, induced by increased TGFβ2 levels, activate TLR4 and exacerbate these signaling pathways in the ONH. This leads us to a fibro-inflammatory hypothesis of TGFβ2 and TLR4 signaling crosstalk between the key ONH supporting cell types leading to glaucomatous damage.

## 4. Fibro-Inflammatory TGFβ2-TLR4 Signaling

Autocrine and paracrine signaling within and between astrocytes, microglia, and LC cells was first proposed over 20 years ago [130,131]. Crosstalk between both TGFβ2 and TLR4 signaling pathways, within and between ONH cells, depends on the ability of the supporting astrocytes, microglia, and LC cells to secrete TGFβ2 and DAMPs, as well as express TLR4 and TGFβ2 receptors. Human ONH astrocytes, microglia, and LC cells all express TLR4 [45,50,132], and all produce TGFβ2 and express TGFβ-receptors [29,50,65,105,133]. This implies that each cell type can respond to increased TGFβ2, increase DAMP production, and respond in both an autocrine and paracrine manner. Here we will outline how the TGFβ2 and TLR4 signaling pathways can communicate with each other to regulate ECM and DAMP production within the ONH.

As we have referenced above, early microglial activation precedes RGC damage and death [31,32], increasing TLR4 expression and cytokine release (Figure 1, #1) [34,35]. These activated microglia, and the increased pro-inflammatory signals, are able to cause the activation of astrocytes in ONH via IL-1α, TNFα, and C1q expression (Figure 1, #2) [44]. These circulating molecules from microglia are necessary and sufficient to induce astrocyte activation and subsequent RGC damage after an initial axon insult [44]. However, RGC damage requires the presence of activated astrocytes also releasing pro-inflammatory cytokines to induce retinal injury and RGC degeneration (Figure 1, #3) [134]. It has been shown that the TLR4-dependent production of pro-inflammatory cytokines is increased in POAG tears [135], AH [136], and in the ONH astrocytes in glaucoma models, indicating astrocyte paracrine and autocrine inflammatory signaling is increased across multiple tissues during glaucoma progression, making astrocyte activation a key instigator of ONH RGC damage [35,44,134].

ONH astrocytes produce the DAMP tenascin-C [67,68,137], a prominent activator of TLR4 that is significantly upregulated in human ONH glaucomatous astrocytes (Figure 1, #4) [67,68,69,75,137]. In primary rat and mouse microglial cell cultures, the DAMP tenascin-C significantly increases IL-6 and TNFα expression levels through TLR4 activation (Figure 1, #2), potentially activating astrocytes through continued paracrine signaling [138,139]. As mentioned previously, the knock-out of tenascin-C was able to inhibit both microgliosis and astrogliosis in a mouse model of glaucoma [116], indicating its ability to act upon both of these cell types through TLR4 signaling. Thus, it is likely that activated microglia and activated astrocytes are able to interact via the paracrine signaling of proinflammatory cytokines and DAMPs. It has also been shown that ONH LC cells also express TLR4 [50], enabling the potential for proinflammatory and DAMP paracrine signaling between all three supporting cell types (Figure 1, #5). Primary ONH LC cells, when stimulated with either TGFβ2 or the DAMP FN+EDA, significantly increase ECM production in a TLR4-dependent manner, indicating autocrine signaling within the monoculture (Figure 1, #6) [56]. Thus, it is likely that similar autocrine signaling is happening within microglia (Figure 1, #7) and astrocyte populations (Figure 1, #8), as well as paracrine signaling between all three population subtypes within the glaucomatous ONH (Figure 1, #9, 10, 11).

In addition, autocrine and paracrine signaling by neurotrophins (NTs) has been implicated in glaucoma disease progression, especially through the loss of such nutrient and growth factor transport through the damaged LC region of the ONH [39,140,141,142]. NTs are a family of nerve-growth factors including nerve growth factor (NGF), brain-derived growth factor (BDNF), glial-derived neurotrophic factor (GDNF), neurotrophin 3 (NT-3), and neurotrophin 4/5 (NT-4/5) [140]. NTs bind to protein tyrosine kinase (Trk) receptors and are highly involved in the peripheral [140] and central nervous immune responses [143]. In the healthy ONH, LC cells and astrocytes produce moderate levels of NTs, and microglia produce negligible levels [130,144]. After acute insults, BDNF is able to exert a neuroprotective phenotype where BDNF injections significantly delay microglial activation post-ON sectioning [145], LC cells and astrocytes increase NT secretions after acute ischemia [131], mimicking the hypoxic microenvironments in the ONH during glaucoma, and activated microglia are known to initially release NGF, NT-4/5, and GDNF [146]. These findings implicate NTs as early responses in protecting the damaged RGCs; however, current hypotheses predict that the constant deprivation of NTs due to the reduced axonal transport greatly contributes to glaucoma disease progression [142]. The local synthesis and retrograde transport of BDNF is significantly reduced after excitotoxic stimuli, leading to retinal degeneration [147], and chronic high IOP induces the loss of BDNF in RGC cell bodies [148]. Importantly, LC cells, astrocytes, and microglia are all able to secrete BDNF and express TrkB, the receptor for BDNF [130,131,149,150]. Not only does this implicate potential autocrine and paracrine signaling between these cell types, but BDNF signaling is thought to slow microglial activation by inhibiting TLR4 downstream signaling, indicating another potential way for NTs to interact between cell types (Figure 1, #12) [149]. The loss of this TLR4 inhibition could be perpetrating the autocrine (Figure 1, #13, 14, 15) and paracrine (Figure 1, #16, 17, 18) feed-forward cycle of TLR4-DAMP activation in all supporting cell types as glaucomatous damage progresses.

Hypothesized signaling crosstalk between microglia, astrocytes, and LC cells in the glaucomatous ONH are represented by the dashed lines in Figure 1 and represent current knowledge gaps in the literature. Further research into this intricate crosstalk is necessary to better understand the glaucomatous pathophysiology.

In the previous sections, we have outlined the hypothesized immune system and NT autocrine and paracrine signaling within and between microglia, astrocytes, and LC cells in the glaucomatous ONH. Both the immune system and NT signaling have direct interactions with TGFβ2 signaling, a predominant contributor to increased ECM deposition in the glaucomatous ONH [56,83,151]. We have previously shown that TGFβ2-dependent ECM production is TLR4-dependent both in primary human TM cells and ONH LC cells [56,83]. TGFβ2 or DAMP exposure from cFN, containing the FN+EDA DAMP, induced significant increases in total FN, collagen-I, and laminin in primary human TM cells, but caused the concurrent blockage of TLR4 signaling by the selective TLR4 inhibitor TAK-242, returning these ECM protein levels back to baseline [83]. A similar phenotype was found in primary human ONH LC cells, where TGFβ2 exposure significantly increased FN, FN+EDA, and collagen-I protein expression, but the concurrent blockage of TLR4 signaling rescued this phenotype back to control levels (Figure 1) [56]. These results implicate that TLR4 signaling is necessary for the TGFβ2-induced ECM production seen in two key structures involved in glaucoma pathophysiology.

The mechanism of TGFβ2 and TLR4 signaling crosstalk has previously been studied in other tissues and disease states through the TGFβ pseudoreceptor BMP and activin membrane-bound inhibitor (BAMBI). It is known that TLR4 activation downregulates BAMBI protein expression in a MyD88-dependent manner via NFκB signaling [98,99,152,153]. In addition, the inhibition of NFκB signaling prevents the downregulation of *Bambi* mRNA after TLR4 activation [99]. When present, BAMBI inhibits TGFβ2 signaling by preventing Smad2/3 phosphorylation, amplifying the Smad7 inhibition of TGFβ2 and acting as a pseudoreceptor, sequestering and inhibiting TGFβR activation and thus Smad3 phosphorylation [82]. We previously reported that the knockdown of *Bambi* in mice induces ocular hypertension and increases ECM production in the TM [154]. In addition, previous studies have implicated a role of BAMBI in other fibrotic diseases [99], and we know astrocytes, microglia, and LC cells can express BAMBI [133,155]. Thus, it seems likely that BAMBI is also an important mediator of TLR4-TGFβ2 signaling crosstalk in the ONH. Each cell can respond to the remodeling of the glaucomatous ONH via the DAMP-dependent activation of TLR4, inducing a pro-fibroinflammatory response. Future studies are needed to fully elucidate the molecular mechanisms of TLR4-TGFβ2 signaling crosstalk in the ONH and the role of each supporting cell type in propagating this response.

## 5. Conclusions

Extensive research has indicated the critical role of both TLR4-immune signaling and TGFβ-induced fibrosis in the glaucomatous ONH in furthering RGC loss. Here, we propose a novel mechanism of TLR4-TGFβ2 signaling crosstalk within and between the supporting cells of the ONH. Elucidating the molecular mechanisms behind this crosstalk in the ONH can hopefully produce more therapeutic targets in treating glaucoma.

## Figures and Tables

**Figure 1 ijms-24-13240-f001:**
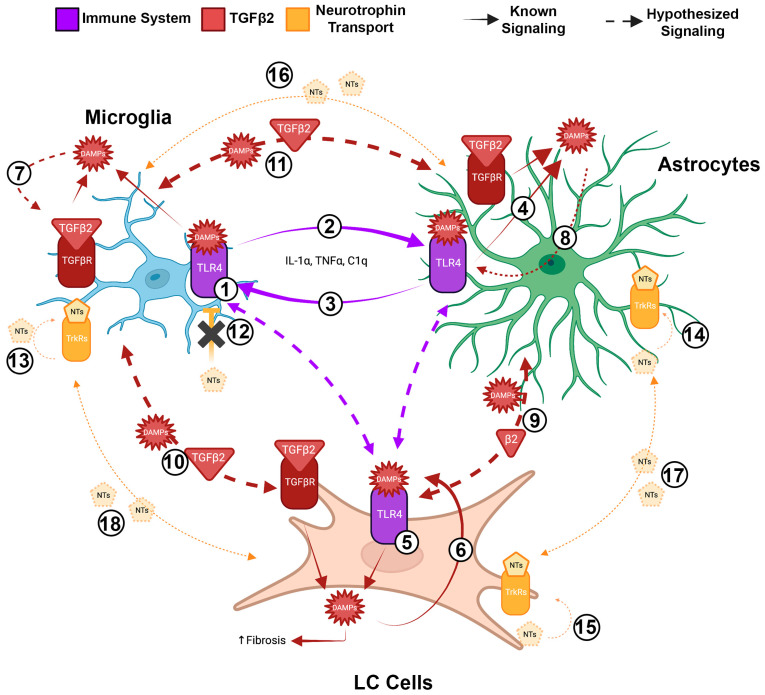
Known and hypothesized autocrine and paracrine signaling in the glaucomatous ONH. Schematic of immune system, TGFβ2, and neurotrophin (NT) autocrine and paracrine signaling. Purple receptors and arrows represent immune system signaling. Red receptors, molecules, and arrows represent TGFβ2 and DAMP signaling. Yellow receptors, molecules, and arrows represent neurotrophin transport signaling. Solid lines represent the interactions shown in previous studies. The dashed lines represent hypothesized autocrine and paracrine signaling pathways that still need to be explored in the glaucomatous ONH. (1) Known early microglial activation. (2) Known microglia → astrocyte paracrine signaling. (3) Known astrocyte → microglia paracrine signaling. (4) Known TLR4-induced production of DAMPs in astrocytes. (5) Known TLR4-induced DAMP production and autocrine signaling in LC cells. (6) Known DAMP–TLR4 autocrine signaling in LC cells. (7) Hypothesized TLR4-induced DAMP production and autocrine signaling in microglia. (8) Hypothesized TLR4-induced DAMP production and autocrine signaling in astrocytes. (9, 10, 11) Hypothesized DAMP-induced paracrine signaling between LC cells, microglia, and astrocytes. (12) Known loss of NT inhibition of TLR4 signaling in microglia. (13, 14, 15) Hypothesized loss of autocrine NT signaling within microglia, astrocytes, and LC cells. (16, 17, 18) Hypothesized loss of paracrine NT signaling between microglia, astrocytes, and LC cells. Created with BioRender.com.

**Table 1 ijms-24-13240-t001:** Glaucomatous fibrotic and immune system responses in the ONH. The ONH-supporting cell types, such as microglia, astrocytes, and LC cells (left column), are each associated with known fibrotic responses (central column) and immune responses (right column) during glaucoma disease progression.

Glaucomatous Fibrotic and Immune Responses in Major ONH Supporting Cell Types		
Supporting Cell Type	Glaucomatous Fibrotic Responses	Glaucomatous Immune Responses
Microglia	- Increased TGFβ2 expression [29] - TGFβ2 treatment upregulates CX3CR1 transcription [59], a potent microglial activator [60]	- Innate immune cells of the CNS [61] - Early activation predicates and correlates with RGC degeneration severity [32] - Increased expression of innate and complement system immune activation genes [34,35] - Inhibition of activation protects RGCs [37]
Astrocytes	- Secrete higher levels of ECM proteins [46] - Primary site of TGFβ2 expression [62] - TGFβ2 treatment significantly increases ECM mRNA and protein expression for FN, COL1, COL4 [63,64,65]	- Undergo astrogliosis: rounded bodies, loss of cell processes [46] - Increased levels of MHC-II, highly involved in the adaptive immune response [66] - Upregulate Tenascin-C, a potent proinflammatory DAMP through TLR4 activation [67,68,69]
LC Cells	- Secrete higher levels ECM [49,54] - TGFβ2 treatment significantly increases ECM transcription and expression for FN, COL1, COL4 [56,57,64,65] - Glaucoma-like insults (mechanical strain, hypoxia) increase ECM expression [51,57]	- TGFβ2-induced ECM production is dependent on functioning TLR4 signaling, a potent activator of the innate immune response [56] - DAMP-induced (FN+EDA) ECM production dependent on TLR4 signaling [56]

## Data Availability

Data sharing not applicable. No new data were created or analyzed in this study.

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
