# Peer review of "The Fibro-Inflammatory Response in the Glaucomatous Optic Nerve Head"

_ijms, 2023, doi:10.3390/ijms241713240_

Round 1

Reviewer 1 Report

This is a well-written and organised review with an interesting topic. A few minor points to consider with regards to the way its written (see highlighted in the attached file). Good effort to create figure 1. Table 1 requires clarification in the legend and labeling. Overall a good review.

Author Response

Thank you for your feedback. To further clarify Table 1 we have reworded the subtitle columns to clarify that we are discussing the role of each supporting cell type in both glaucomatous fibrosis and immune response as well as bolded the subtitles to indicate that it is a label. We have also expanded on the suggested Table 1 figure legend to further emphasize this distinction. We changed the term “I” to “we” in line 313. We un-italicized the word “necessary” in line 416. All document changes are highlighted in yellow.

Reviewer 2 Report

  1. Please note that references should be marked before a period.
  2. The author proposes a new mechanism of glaucoma injury in the manuscript, but this seems to be only a hypothesis. Authors need to give possible experimental schemes or routes, so that future researchers can confirm the rationality and authenticity of the proposed mechanism.
  3. use gender instead ‘sex’ in line 21.

Minor editing of English language required

Author Response

Thank you for your feedback. We have fixed our reference errors to place them before the period. We have emphasized the areas for further potential studies, focusing on the signaling crosstalk, in lines 397-400. According to guidelines from the Journal of Applied Physiology (doi:10.1152/japplphysiol.00376.2005) and the NIH (https://orwh.od.nih.gov/sex-gender), the term ‘sex’ instead of ‘gender’ in line 26 is appropriate as we are referring to the biology of human subjects across multiple genomic and population based studies and not referencing self-identity or societal populations. All document changes are highlighted in yellow.

Reviewer 3 Report

The review article titled “ The Fibro-Inflammatory Response in the Glaucomatous Optic 2 Nerve Head” is talking about fibrotic and inflammatory aspects of glaucoma damage in the optic nerve head.  The authors have reviewed known inflammatory and fibrotic changes occurring in the ONH. In addition, they described a novel mechanism of toll-like receptor 4 (TLR4) and transforming growth factor beta-2 (TGFβ2) signaling crosstalk in the cells of the ONH that contribute to glaucomatous damage. Understanding molecular signaling within and between the cells of the ONH can help identify new drug targets and therapeutics. This is a well-written review article, though it has some limitations.

The point-wise comments are as follows.

·         There are four major glaucoma type,  Open-angle glaucoma, Angle-closure glaucoma, also called closed-angle glaucoma, Congenital glaucoma, and Secondary glaucoma. Does the risk factor and symptoms are the same in the 4 kinds of glaucoma?

·         Does all the known glaucoma type, exert the same damage to the ONH?

·         What is the difference or similarities of Fibro-Inflammatory responses in all 4 types of glaucoma?

1.       

Author Response

Thank you for your feedback. We have added a sentence in lines 21-26 describing how the glaucomas are a heterogenous group of neuropathies defined by the damage to the ONH, RGC loss, and well-characterized visual field loss. The molecular pathology has not been fully elucidated for any subtype of glaucoma, although given many common risk factors (ex. Elevated IOP) we would anticipate a similar molecular and cellular response in the ONH cells. Here we also indicated that we are focusing on the most common form of glaucoma, primary open angle glaucoma, in this review. All document changes are highlighted in yellow.